# Management of Hemorrhagic Shock: Physiology Approach, Timing and Strategies

**DOI:** 10.3390/jcm12010260

**Published:** 2022-12-29

**Authors:** Fabrizio G. Bonanno

**Affiliations:** Department of Surgery, Polokwane Provincial Hospital, Cnr Hospital & Dorp Street, Polokwane 0700, South Africa; f.g.bonanno@gmail.com

**Keywords:** hemorrhagic shock, fluid load test, hypotensive resuscitation, exsanguination, titrated-to-response anesthesia, damage control surgery

## Abstract

Hemorrhagic shock (HS) management is based on a timely, rapid, definitive source control of bleeding/s and on blood loss replacement. Stopping the hemorrhage from progressing from any named and visible vessel is the main stem fundamental praxis of efficacy and effectiveness and an essential, obligatory, life-saving step. Blood loss replacement serves the purpose of preventing ischemia/reperfusion toxemia and optimizing tissue oxygenation and microcirculation dynamics. The “physiological classification of HS” dictates the timely management and suits the ‘titrated hypotensive resuscitation’ tactics and the ‘damage control surgery’ strategy. In any hypotensive but not yet critical shock, the body’s response to a fluid load test determines the cut-off point between compensation and progression between the time for adopting conservative treatment and preparing for surgery or rushing to the theater for rapid bleeding source control. Up to 20% of the total blood volume is given to refill the unstressed venous return volume. In any critical level of shock where, ab initio, the patient manifests signs indicating critical physiology and impending cardiac arrest or cardiovascular accident, the balance between the life-saving reflexes stretched to the maximum and the insufficient distal perfusion (blood, oxygen, and substrates) remains in a liable and delicate equilibrium, susceptible to any minimal change or interfering variable. In a cardiac arrest by exsanguination, the core of the physiological issue remains the rapid restoration of a sufficient venous return, allowing the heart to pump it back into systemic circulation either by open massage via sternotomy or anterolateral thoracotomy or spontaneously after aorta clamping in the chest or in the abdomen at the epigastrium under extracorporeal resuscitation and induced hypothermia. This is the only way to prevent ischemic damage to the brain and the heart. This is accomplishable rapidly and efficiently only by a direct approach, which is a crush laparotomy if the bleeding is coming from an abdominal +/− lower limb site or rapid sternotomy/anterolateral thoracotomy if the bleeding is coming from a chest +/− upper limbs site. Without first stopping the bleeding and refilling the heart, any further exercise is doomed to failure. Direct source control via laparotomy/thoracotomy, with the concomitant or soon following venous refilling, are the two essential, initial life-saving steps.

## 1. Introduction

A “physiological classification” of hemorrhagic shock has been proposed since 2010, based conceptually upon the right definition of shock, the body’s natural hemostatic mechanisms, the new developments in the microcirculation/arteriolar system, the concept of ‘cardiac/circulatory reserve’, and basic physiological considerations, such as the significance of fluid-resistant hypotension. It is a clinical classification that fits well with the optimal timing of management and the implementation of damage control surgery (DCS) and titrated hypotensive resuscitation (THR) [1] (Figure 1 and Figure 2). Relevantly, it introduces some order into the chaos of the literature, with neologisms such as lethal triad and damage control resuscitation, which have yielded no practical benefits. 

At present, only in vivo peripheral microcirculation visualization and continuous cardiac output (CO) monitoring through trans-thoracic echocardiography (TTE) can add to clinical assessment in predicting the degree of compensation or decompensation. The integration with tissue hypoxia’s evaluation, such as ScvO_2_ measurement, would be welcome.

Artificial intelligence is what can help in the prediction of compensation, the capacity of compensation, and the duration of compensation. Until artificial intelligence becomes more refined and sophisticated in measuring real-time shock dynamics in its macro- and microcirculatory component, the management and timing of intervention remain based on the clinical assessment of signs correlating to the degree of the patient’s current hemodynamic derangement [2].

To date, the research is ongoing. A continuous, real-time arterial waveform analysis (AWFA) can provide early hemorrhage detection and an indication of the onset of overt shock before the standard vital signs and allow an early distinction between “poor” compensators (i.e., those with relatively low tolerance to blood loss) and “good” compensators [3]. Advances in the measurement of tissue oxygenation have recently been achieved with resonance Raman spectroscopy, which appears to be as fast and accurate as standard conventional methods such as central venous hemoglobin oxygen saturation (ScvO_2_) [4]. There are no methods yet to measure the cardiac/circulatory reserve and tissue oxygenation in real time easily and accurately enough for perfectly timed decision making. 

### 1.1. Cardiac Reserve and Fluid Load Test

No critical care situation, whether cardiogenic shock, septic shock, or HS, can disregard three fundamental physiological concepts: the fluid responsiveness test, the cardiac/circulatory reserve, and microcirculation, specifically, the arteriolar system on which the distal tissue perfusion depends [1,2].

The cardiac reserve, which in the case of an HS is better defined by the term “circulatory reserve”, is the maximum quantity of blood that can be pumped above the baseline normal level during exercise or for the compensation of deficits within physiological limits. The lower the myocardial functional reserve is or becomes, i.e., the healthy part of the myocardium capable of responding to venous return variations during diastole with an increase in contractility according to the Frank–Starling law, the higher the chances of a shock following the failure of the pump. This occurs because decreased compliance leads to increased left/right ventricular end-diastolic pressures, causing increased wall tension and increased oxygen consumption (VO_2_), which in turn decrease the CO [5,6]. As put in words by Convertino, any compensatory physiologic mechanism (e.g., tachycardia and vasoconstriction) has a finite maximal response to any specific stress (e.g., hemorrhage). The compensation “reserve” can also be defined as the difference between the maximal response and the baseline state [7].

The clinical observation of the shock is what is commonly used, measuring the systolic pressure as a direct variable of CO and correlating it with the general clinical picture.

In the absence of myocardial or valves disease or dysrhythmia, or hypertension in the elderly, especially with the co-presence of a TBI, or of interfering drugs, the SBP response to a fluid load in patients with normal circulatory/cardiac reserve can be relied upon and accepted as a rough guide to the CO. If after 20–30 min of a fluid load, the SBP and its indirectly related tachycardia have not improved and maintained stability, it can be assumed that the current cardiac/circulatory reserve and its directly related CO are not sufficient to keep the patient hemodynamically stable. 

Patients who respond with the normalization of SBP and a reverse trend of tachycardia are considered “responders”.

If the CO and its direct variables—systolic pressure and tachycardia—have not improved and maintained stability over a 20 min observation span, it can be assumed the current body reserve capacity is not sufficient to keep the patient hemodynamically stable. This situation defines a non-responder scenario [1]. 

A CO direct serial response to fluid load would conveniently be the most reliable parameter we could use to measure the cardiac/circulatory reserve, due to its direct relationship with the venous return.

CO can be directly and continuously measured at the bedside with a trans-thoracic echocardiogram (TTE) following fluid challenge/s in 5–10 min. Patients should undergo repeated regular echocardiography monitoring for at least a 20 min period, and up to 30 min in the presence of kidney failure, to exclude transient response [2]. 

CO measured by TTE always precedes the changes in tachycardia and blood pressure to follow.

TTE needs skills and experience to be very accurate but remains the best available means to monitor directly CO in a fast and prompt way. 

A change in the pressure gradient of venous return, defined as the difference between the mean systemic filling pressure (Pmsf) and central venous pressure (CVP) following a fluid challenge, is seen in responders but not in non-responders. In the non-responders, the increase in Pmsf translates solely into an increase in CVP [8]. In those that respond, the maximal change in CO is seen 1 min after the completion of the fluid challenge. 

Pmsf is the measurement of the pressure when there is no flow in the vessels like in a circulatory arrest. Intravascular volumes are well divided into “stressing volume” and “un-stressing volume” with reference to the vessels’ wall tension [9]. The un-stressing volume fills the vessels but does not generate any pressure. The stressing volume causes a stretch in the vessel walls and increases the pressure within the vessels. If an effective fluid challenge is given, it will, at least transiently, increase the stressing volume and cause a rise in Pmsf. This increases cardiac preload, which ultimately increases CO in the preload-responsive patients according to the Frank–Starling principle.

If a fluid challenge is given, which is effective in significantly increasing Pmsf, and no subsequent increase in CO is seen, the patient is labeled as non-responsive. In normovolemic conditions, such as in septic shock, if the patient is fluid responsive, i.e., has a functional arteriolar system, an effective fluid challenge will result in a significant increase of more than 10% in the stroke volume or CO witnessed consistently with repeated examinations for 20 min [10,11].

The increase in CO is a transient response: a return to baseline values is seen 10 min after fluid administration [12]. The optimal amount of fluid challenge test in septic shock is ≥4 mL/kg, the amount incidentally used in repeated doses for hypotensive resuscitation in HS [13]. There is a substantial difference between a fluid challenge in an inflammatory or distributive shock and in a hemorrhagic one, and that is normovolemia in one and hypovolemia in the other one.

Despite the differences in dynamics between the septic shock, where the fluid challenge test has been extensively studied, and the HS, in terms of volemia, the same principle and rationale, with due modification, can be applied and reliably standardized in the management of HS.

In acute hypovolemia, like in the hypotensive HS, fluid challenge should be given in volumes sufficient enough to bring first the V/P ratio to a less negative level than the one occurring once hypotension sets in, while at the same time, keeping the ratio not close to the unity in case this would counteract the physiological hemostatic mechanisms.

Due to hypovolemia not being present in the septic shock situation, only by increasing first the mean systemic filling pressure (Pmsf) it is possible to know if the challenge test can be relied upon in the appraisal of an HS. The aim of a fluid challenge in a progressing hypotensive shock is to determine whether the blood loss has been controlled by the physiological mechanisms and a functional arteriolar system or is progressing and we are dealing with an uncontrolled hemorrhage.

Restoring the blood volume loss that had produced hypotension, which in an adult is at least 20% of TBV, is therefore the first step.

More than that amount in an adult, for example, 1.5 L, which equals approximately 30% of TBV and is the amount that if lost gives consistent hypotension, might actually increase bleeding by counteracting the physiological hemostatic mechanisms [1].

Consequently, it can be seen as a reasonable and acceptable tactic to perform the following order of steps. Give a 500 mL bolus of fluids in 5 min, and repeat it again within the following 510 min, if no hemodynamic changes are seen, specifically, the normalization of SBP and inverse trend of tachycardia [2]. Observe the possible meaningful effects on CO or BP. Wait a further 10–20 min to see if the increase in SBP remains stable.

This empiric dosage has been indirectly confirmed by the experience in normovolemic septic shock [13] and by a study where increased mortality was seen with a >500 mL bolus prn given to a group of patients with blunt trauma without prehospital hypotension, compared to the same bolus given to a group with prehospital hypotension [14]. The amount of the bolus happens to be 7 mL/kg BW and 10% of TBV.

The test fluid should be one supporting at this stage the macro-circulation; if the microcirculation flow dynamics is also facilitated, the better.

HTS would be the optimal fluid in HS combined with TBI, but one cannot be given more than 375 mL/h. If HTS is used, it should be followed by another fluid, as one cannot be given not more than 250 mL/h (max 375 mL in 1.5 h) to maximize its merits and diminish its drawbacks on bleeding by interfering with coagulation and inducing a possible increase in pressure as a direct hemodynamic effect [15,16,17,18].

Practically, any colloid or crystalloid can be used as a fluid for the load test.

At this stage of diagnostic limbo between HS clinical stages 2 and 3, precious plasma blood and blood components should be spared.

### 1.2. Management Options of Compensated Stable Shock

Moderate shock levels respond well to crystalloids and PRBC iv once the fluid test has ascertained a functioning arteriolar system. Basic CCT can be used as coagulation screening. There is no need for routine VHA tests in compensated shock. Blood transfusion should be carried out soon after source control until a level judged as satisfactory. Hb ≥ 7 g/dL with Hct ≥ 22% in healthy patients and ≥ 9 mg/dL with Hct ≥ 27% in cardiac patients. SvO2 minimal level of 70%, an NBE < 4 meq/L, and a LA < 4 mmol/L in absence of infection are acceptable end-points [19].

Patients do not require immediate or rapid surgery but can be investigated before surgery with a CT scan or interventional radiology and considered for conservative management. The principle of management remains the same: the earlier the source control, the better outcomes, particularly for the inevitable ischemia-reperfusion toxemia always lurking and proportional to the duration and to the level of ischemia [20,21,22,23,24,25].

In a compensated stabilized patient, a maximum period longer than 4–6 h has been found safe before source control, as seen from experience in rural surgery or in countries with developing health systems or warfare scenarios, where a compensated HS has been seen to survive even with >6 h’ lack of treatment [26].

These observations were crucial for the adoption of non-operative-management (NOM) of trauma in the not actively bleeding patients with hemorrhage, which is tamponed by parenchymal tissues or by organs anatomical boundaries, e.g., the liver [27]. The six-hour limit is the time taken by the post-traumatic inflammatory response in trauma to be fully blown. That limit is also the safest time within which an inflammatory response to hemorrhage in the abdomen is elicited with palpation.

Stable patients with penetrating injuries have withstood delays of intervention up to two hours without complications [28].

NOM decision depends on the general conditions of the patient, hemodynamic stability, absent abdominal tenderness, no current bleeding, and an identified contained bleeding source in an anatomical boundary [26,27].

Patients who arrive at a surgical facility after 6–8 h in stable hemodynamics, who do not have foreign bodies adjacent to vital structures or peritonitis, can undergo nonoperative management. Surveillance depends on the type of the site of injury and varies from 2 to 7–10 days [29].

Investigations are allowed only in not-hypotensive, compensated, mild-to-moderate HS, and should aim only to identify the origin of the bleeding/s and concomitant pathologies worthy or essential to be picked up before surgery.

In elderly hypertensive with TBI, any drop in pressure within the usual normal range should be seen as mere hypotension, and an SBP of 110–120 should be seen as shock [30,31,32,33].

The shock index cannot be relied upon in the context of trauma where pain, inflammatory response, hypoxemia, or hypercapnia can raise the pulse, but it is a useful parameter in non-traumatic hemorrhage [34].

## 2. Hypotensive, Unstable, Decompensated, Progressing Shock—Stage III

Once there is a no response to the fluid load test or only a transient response not longer than 2030 min, the time the rapid hemodynamic compensation reflexes fully set in [7], the patient can be deemed in an unstable advanced shock, decompensated for the failure or dysfunction of the arteriolar braking system, with bleeding progressing. No investigation should be entertained in the presence of progressive shock not responding to fluid load.

Subsequent management depends on whether the bleeding faced is outdoors or indoors, and whether it is arterial, venous, or a mix/unknown [Figure 3].

### 2.1. Management Outdoors

#### 2.1.1. Titrated Hypotensive Resuscitation

If it is known or highly likely that the bleeding source is arterial, ‘titrated hypotensive resuscitation’ (THR) tactics should be used.

The tactic of THR was developed from historical observations of the beneficial effects of hypotension in the spontaneous decrease or stoppage of bleeding. Normal–high pressures, before source control, are detrimental in counteracting the physiological mechanisms stopping bleeding (hypotension, arterial spasm and vessel retraction in arterial bleeding, and clot formation in venous bleeding) [1,5,30,35,36,37,38,39,40,41,42,43,44,45,46].

HR should not be used in children <15–16 years of age, in hypertensive elderly, where any drop in pressure within a normal range should be seen as mere hypotension, and in pregnant people where any hypotension, indicating >35% TBV loss, is an ominous sign of life-threatening physiology.

An ascertained venous source of bleeding is another contraindication to HR.

The tactic serves the purpose of keeping the patient’s consciousness level and mean perfusion pressures at sufficient enough levels to decrease the impact of IRT and prevent death from CA by insufficient venous return in arterial bleedings.

By its very definition, it can work only within the dynamic of arterial bleeding. If applied in venous or prevalently venous bleeding, HR can, in fact, accelerate physiological deterioration, as it decreases the amount of fluid that is possibly needed to fill all portions of the venous system that are intact, decreasing the venous return to the heart, and hastening the blood loss.

If HR does not reach its aims of keeping minimal safe MAP in an unstable patient, then the patient is escalating to exsanguination and entering phase IV or a critical HS level of impending CA.

It is in the subgroup of temporary fluid responders keeping on destabilizing repeatedly after short-term stabilization and in a known arterial or presumed as such bleeding that THR finds its optimal and most useful application, aiming to keep consciousness level and a minimally sufficient MAP while rushing to the nearest surgical facility for a crush GA/surgery.

The reason why hypotensive resuscitation has not clearly shown benefits on morbidity and mortality is in the lack of uniformity of the meta-analysis as far as indications, physiological derangement, the origin of the bleeding, and the timing of iatrogenic vasoconstricting support [43,44,45].

Without the right pathophysiological interpretation, a dynamic physiological classification, and the real-time reading of the patient, HR can turn out to be a damaging gimmick [46].

#### 2.1.2. Which Amount of Fluid to Use for HR Tactics?

Clinical experience in civilian and military settings has shown the benefits and safety of boluses of 250 mL of fluids, whether crystalloids, colloids, or hypertonic saline (HTS) at different concentrations, as effective and safe initial management for patients with HS. The amount corresponds approximately to 5% of TBV. If HTS is used, it should be followed by another fluid as one cannot be given not more than 250 mL/h (max 375 mL in 1.5 h) to maximize its merits and diminish its drawbacks on bleeding by interfering with coagulation and a possible increase in pressure as a direct hemodynamic effect [15,16].

Crystalloids are probably better to be avoided due to the proven deleterious effects on hypotensive non-compensated HS [1].

#### 2.1.3. Which Fluid Is Best?

The fluid for THR should be the same fluid used as the test load, and the one we would use as a bridge infusion until source control.

All types of fluids have been tested.

There is a substantial difference between the fluid used in the load test, a test for the arteriolar gate system efficacy, aiming mainly to boost macro-circulation volumes, and a fluid needed for maintaining minimally sufficient volumes and pressures in the macro-circulation as well as addressing and optimizing microcirculation function.

The type and quantity of the fluid used in THR are relevant as each type has its advantages and drawbacks. Microcirculation flow is sustained by two preponderant factors: pressure, which depends indirectly on the upstream one, and the Poiseuille law of laminar flow, where viscosity is the most relevant variable and incidentally the most affected by crystalloid administration.

There is overwhelming evidence of the importance of maintaining microcirculation function, with the aim to optimize perfusion in HS, by using fluids with specific characteristics and compositions, particularly viscosity, the most crucial factor of the Poiseuille equation ruling micro-circulation flow, more than colloid osmotic and oncotic properties. In experimental conditions with small animals, hypertonic saline (HTS) or synthetic colloid combinations (hyper-osmotic, hyper-viscous solutions such as HTS, RL, or Dextran 70 combined with alginates, or polyethylene glycol (PEG) conjugated albumin solutions) have been shown to represent an optimal choice for microcirculation dynamics [47,48,49,50].

Blood has an irreplaceable property and function from micro-hemodynamics and oxygen-transport end-points [50,51,52,53].

Colloids are superior to crystalloids as far as hemodynamic stabilization for two main reasons: (i) the increased oncotic/colloid osmotic property is particularly beneficial in the microcirculation, and (ii) the smaller required volumes are an advantage outdoors for preventing the continuation or recurrence of bleeding from high pressures/volumes in the macro-circulation. These properties, however, are offset by the interference with coagulation as well as with immune response and renal function by many of the standard colloids.

#### 2.1.4. What to do if THR Fails?

Here, is where a therapeutic window is available to support an arterial system about to collapse for insufficient ADH production by adding iatrogenic ADH. The rationale is to strengthen the arteriolar braking system until rushing to the nearest surgical facility. ADH is preferred to NA in the preoperative period unless the patient circulation arrests in the progression; then, the side effect of NA loses ground in favor of its combined vasoconstrictor inotropic and bradycardic actions. In postoperative ICU, NA can substitute or be added to ADH if necessary, in safer circumstances and under monitoring [54,55,56,57,58,59,60,61].

The compensation of hemorrhagic shock occurs by reflex sympathetic-mediated arteriolar vasoconstriction with catecholamines acting on α1 receptors. At some stage, hypo-reactivity to endogenous catecholamines sets in, signaling decompensation, which eventually becomes paralysis with a lack of responsiveness even to exogenous vasoconstrictors, signaling irreversible shock. This is very probably due to hypoxia hitting the arteriolar endothelial cells [24].

NE should be added to AVP if the latter is ineffective and must be discontinued before ADH [62].

Overall, though, besides the sparing of blood requirements [63], there is a lack of studies showing benefits on morbidity and mortality with vasoconstrictors [50,64].

The main reason for this failure is the mistiming of drug delivery. Vasoconstrictors should be given when HR fails in the critical lag between stage III and stage IV, with the aim to prevent patient exsanguination at the expense of distal microcirculation. An amount of ischemia-reperfusion toxemia (IRT) is to be expected in survivors.

If HR does not work even with iatrogenic vasoconstrictors, the patient is going quickly to stage IV.

#### 2.1.5. What to Do If the Bleeding Outdoors Is Known to Be Venous?

Two strategies are feasible.

One is to give blood or its components or fluids in an amount totally equivalent of up to 40% of TBV. If the patient stabilizes, they are then brought to a surgical facility under the “keep the veins open” modality solely. In the presence of known ischemic heart disease (IHD) or cerebrovascular disease (CVD), blood would be the fluid to use.

Blood or blood components are continuously transfused via a central venous line until rapid source control. As an alternative, two 14-16 G peripheral cannulas can be inserted. Though blood or its substitutes would be lost in the solution of the continuity of the macro-circulation, it would perfuse the brain and heart up to the control of the hemorrhage. Availability and costs bind its use to a strict protocol for its use outdoors.

A pitfall is to transfuse into a vein feeding the bleeding spot.

If instead the patient does not become stabilized and does not respond to fluid therapy, they are entering stage IV.

In this scenario, a continuous transfusion and a rush to a surgical facility would then be the choice.

The second option is to “scoop and run” towards the nearest surgical facility with no treatment on the spot other than venous cannulations.

In a landmark study on hypotensive patients with penetrating injuries to the torso randomized in a group who underwent standard fluid resuscitation according to the classic longstanding guidelines of ATLS and a group who received only intravenous cannulation but were not transfused, the latter scored better in survival, morbidity, and hospital stay [65].

The fundamental message of Bickell’s study has been predictably confirmed with a clear association between a long time on the scene, longer than the time required to scoop the patient and run, with mortality [66,67].

The above strategy can also be applied to bleedings, known to be mixed venous/arterial ones.

#### 2.1.6. What Fluid to Give in Prehospital Settings If Continuous Transfusion Is the Choice?

Blood loss replenishment is the complementary management to the mandatory, essential, source control step. Since Claude Bernard introduced the concept of “physiological homeostasis”, by mere basic logic and common sense, if it is the loss of blood that occurs in hemorrhage, it is only blood or blood components that should be given for replenishment [68,69,70,71,72]. Blood loss replenishment serves the purpose to prevent or attenuate the obligatory ischemia-reperfusion phenomenon, which is proportional to the level and length of the hypoxemia period. Blood is the natural occupant of the vascular system and as such is responsible for the vascular basic tone and dynamic pressures and for heart volumes and pressures. Besides the characterizing red blood cells carrying oxygen, blood carries nutrients, inflammatory and immune cells, hormones, and coagulation cells and factors. Whole blood is in principle the best and ideal replenishment fluid, especially 24 h fresh or walking-bank whole blood. Alternatively, RBC and plasma as the second choice and RBC and Ringer Lactate as the third choice are commonly used. Blood remains the not-substitutable, natural, ontogenetically programmed, essential-for-living oxygen-carrying agent [73,74]. It carries also other essential gasses such as CO_2_ and NO. Blood’s essential role is manifested also in its capacity to maintain flow in the microcirculation, due to its perfect viscosity [75,76], the most relevant variable in the Bernoulli equation. A further, non-substitutable, role of fresh whole blood is the capacity of red blood cells together with nitric oxide (NO) in regulating microcirculation [77,78,79,80,81]. Finally, whole blood accelerates clot formation [82], while synthetic fluids prevent or impair clot formation or stabilization. The problem with the universal use of blood outdoors remains its availability.

Fresh whole blood, walking bank blood, or cold-stored low titer 0 whole blood (LT0WB) obviates the need for separate component transfusion. Whole blood compares with blood components in a 1:1:1 balance of platelets/packed red cells/plasma as far as safety and hemodynamic performance [83,84,85,86,87,88,89,90]. The issue is still a working progress [91]. It is not clear if blood products consumption would be decreased with WB transfusion at a lesser amount [92]. Walking bank or fresh whole blood is ideal, due to the preserved oxygen carrying capacity and release, but cold-stored low titer group 0 (LT0WB) keeps platelets healthier and viable when stored up two weeks [93,94]. 

There is no clear evidence, however, of advantages in survival by giving whole blood or blood components in hypotensive patients in prehospital settings [90,91,92,95,96]. Only one study apparently showed an improvement of hemodynamics and a reduction in early 6 h mortality with LT0WB in theatre compared to no blood (crystalloids or no fluids) in patients with advanced shock, but with no improvement on mortality on the scene. That study was actually biased by the negative effects of crystalloids and by the overall higher gravity of pathology in the no-LT0WB group [97].

The absence of the advantages of pre-hospital settings confirms that the fastest source control is the primary factor for survival mortality and morbidity, and the complementary role of blood fluid replenishment is mainly for avoiding or preventing IRT and SIR. Without source control, as easy and banal as it can be, no massive transfusion would save a patient with hypotensive uncontrolled progressive hemorrhage [1,2]. HS is present in up to 40% of trauma [98]; in 12–34% of trauma and hemorrhage, deaths are preventable by advanced intervention on the scene within one hour from a surgical facility [99,100]. Most deaths of HS not exsanguinating on the scene die within 2 h of the bleeding onset [101]. An association between a longer time on the scene and mortality has been seen in hypotensive severe HS [102]. This means the categorization of the patient according to their physiological status and circulatory reserve on the scene is vital for the outcome of advanced HS [1,2].

Plasma is the prototype of colloids, being a natural colloid with the right properties to sustain microcirculation. Normally, FFP is beneficial in patients massively bleeding as a substitute for multiple coagulation factor deficiencies up to 20–30 mL/kg BW, particularly in microvascular bleeding, i.e., oozing, in massive transfusion, disseminated intravascular coagulation (INR > 2), severe burns, overzealous oral anti-coagulant management (warfarin), and liver disease coagulopathy.

Prehospital plasma, likewise, prehospital blood, whether given as thawed FFP (AB universal donor), pre-thawed liquid (type-specific), or freeze-dried prehospital plasma, has not shown benefits on HS mortality [90,92,103,104,105,106,107].

Unmatched liquid plasma or freeze-dried plasma nowadays has replaced ABO-matched fresh frozen plasma, allowing early administration without logistic problems. There is, however, unconvincing evidence that prehospital plasma decreases mortality compared to no plasma or other fluid transfusions [92,103,106].

Its freeze-dried and liquid formulations have solved the problems of conservation, storage, and simultaneous hemotherapy when given together with RBC and platelets in a 1.1:1 ratio [108,109,110,111].

A low-volume spray-dried modification of FDP reconstituted to one-third of its original volume without compromising the procoagulant properties in vivo, being more stable, and also maintaining hyper-oncotic and hyper-osmotic properties has been successfully tested in animal models [112,113,114,115].

Its advantages, in concomitance with blood transfusion too, have been seen only in blunt injuries, traumatic brain injury, and transport time greater than 20 min [116,117,118]. Why in blunt injuries only?

In normovolemic inflammatory shock, e.g., septic shock, plasma appears to give benefits in preventing endothelial damage [119,120], with consequences in the microcirculation dynamics [121,122,123]. In some blunt trauma, for example, in a predominant orthopedic trauma with scarce blood loss, a post-traumatic inflammatory response ensues. In these scenarios, the inflammatory cascade has more chance of being retained inside circulation, damaging the endothelium and microcirculation. This observation may well explain why plasma has benefits in blunt injury when added to blood.

The advantages of TBI may well be explained by decreasing the entity of coagulopathy and, therefore, the threshold of intervention in mild or moderate TBI.

The association with blood transfusion may be the contributing factor to benefits in delayed transport, probably due to the blood-enhanced clot formation [82].

Plasma remains harmful when over-transfused in patients who are not actively bleeding as it causes transfusion-associated circulatory overload and transfusion-related acute lung injury [124,125]. Circulatory overload, acute lung injury, and the dilution of RBC, fibrinogen, and platelets are common side effects [126,127,128].

There is also no evidence that it enhances clot formation and increases fibrinogen, despite containing all coagulation factors (including fibrinogen at >70% of normal levels), resulting, in the end, in a wasted resource [129].

No benefits on survival have been seen in two studies with RBCs and plasma due to bias in the timely administration of the various components and in the incorrect delineation of inclusion criteria and confounding factors such as hypotension, intubation, mechanical ventilation, hemodynamic derangement level, and varied pre-hospital infusion management [106,130].

Its utility in HS is controversial and limited, as well as ineffective, deleterious, and a waste, except when used as fluid for HR or in the specific scenarios of burns, concomitant anticoagulants, DIC, and liver disease.

Platelets are transfused in a balanced 1:1:1 ratio together with plasma and red blood cells. It is nowadays impossible to foresee the efficacy of platelets transfused. Platelet dysfunction is a problem and cannot be reliably identified by tests. A normal number does not exclude coagulopathy, nor does transfusion guarantee the restoration of platelet counts or function. Moreover, the dilution in plasma or synthetic fluids affects their efficacy in clot formation [129,131,132,133].

Aspirin and the other anti-platelets drugs do not affect outcomes in the hemorrhage of solid organs [134], nor do warfarin or the OAC affect outcomes in bleeding patients with no head injury [135,136]. All evidence comes towards the confirmation that coagulopathy does not affect outcomes and is a much less important dynamic in shock, and, anyway, it should not be considered a criterion for damage control surgery [137].

An excellent review of the management of patients with TBI on anticoagulants is available [138].

No procoagulant factor can affect outcomes in HS in the absence of a concomitant TBI where it increases the threshold for intervention in mild or moderate head injury; other advantages are to be found in decreasing the needs of blood or blood products in the postoperative period in subjects with deficiencies and in increasing the options of NOM [137].

Cryoprecipitates are given after the stoppage of bleeding in patients requiring specific factors to restore their coagulation profile; when a test of fibrinogen activity indicates fibrinolysis; when the fibrinogen concentration is less than 80–100 mg/dL in the presence of bleeding; in patients with hemophilia (von Willebrand disease) that cannot be treated with available desmopressin and VWF/FVIII concentrate.

No procoagulant factor or anti-fibrinolytic agent can stand the preponderant hemodynamic forces of a pumping heart and vascular pressures behind bleeding [137].

Tranexamic acid should be given only in HS with a TBI of mild or moderate entity [137,139,140,141,142], whether in HS or not, in order to prevent the increase in hematoma sizes to a threshold for intervention.

The claims that the fixed protocol of FFP/Plts/RBC in a 1:1:1 ratio increases survival are unfounded and biased by the effect of the RBC component transfusion on survival and clot formation rather than from addressing an assumed or factual coagulopathy *per se* [126,143].

Crystalloids, as a frontline immediate management fluid, even if given with RBC and plasma, can only worsen bleeding by hemodilution of coagulation factors, dislodgement of clots, vasodilation, and increase of pressure [143,144]. Bickell et al., in a landmark study on patients with penetrating trunk injury, hypotension, and uncontrolled vascular injury, noted that if no fluids in a standard fashion were given in a prehospital setting before theater, survival was increased, complications decreased, and hospital stays shortened compared to standard fluid resuscitation [65]. Aggressive crystalloid-based resuscitation strategies were also associated with cardiac and pulmonary complications, coagulation disturbances, and immunological and inflammatory mediator dysfunction [145], and were found to correlate with a higher rate of coagulopathy, organ failure, and sepsis rate when compared with lesser amounts of fluids [146].

Only recently, ATLS has scaled down to 1 L from the previous 2 L as the suggested amount of crystalloids for the initial management of HS [147]. Interestingly, a study of <500 mL of crystalloids vs. >500 mL as the initial fluid management in normotensive and hypotensive shock patients did not affect the hypotensive group of patients but increased mortality only in the normotensive group [14].

The loading fluid test of 2 L of crystalloids, previously recommended by ATLS^®^, was *de facto* anti-physiological and deleterious, especially when indiscriminately implemented, and did not bring an increase in survival [148] but an increase in mortality and postoperative complications when compared to no-fluids or fluid-restriction resuscitation [65].

#### 2.1.7. *Resumé* of Fluid Strategy in Progressing HS Stage III

Whole blood; 1:1:1 plasma, RBCs, and platelets; 1:1 plasma and RBCs; reconstituted DP (not available presently), liquid or thawed plasma, alone or RBCs alone; Hextend; RL or Plasma-Lyte, in order, would be the logical priority of utilization in progressing shock in base to specific necessity or availability.

Common sense and reading the patient’s actual physiological level are paramount [149].

In HS level III, blood or its components are better spared at the beginning of the attempt to stabilize an unstably stable patient at the impact, as long as THR works and oxygen is well titrated.

All microcirculation studies have affirmed that viscosity and the other variables are more crucial than the oxygen-carrying capacity of erythrocytes in keeping tissues oxygenated enough [73,74,75,76]. Moreover, there is not enough evidence that prehospital blood or its components increase survival [92,103,106,137].

Therefore, if viscosity can be maintained with an apt fluid, oxygen is well titrated, and THR works, less expensive fluids can keep going until reaching the fastest source control in a surgical facility. Blood or its essential components can then be spared and utilized better in a more targeted mode, specifically if the patient at stage III was known to have pre-existing ischemic heart disease or cerebrovascular disease [19], situations where the oxygen-carrying capacity of blood would make a difference to resuscitation. In this circumstance, to use blood in quantified amounts during THR tactics is reasonable and justified.

When instead a continuous transfusion is necessary, due to preponderant venous or mixed bleeding, the limited blood availability can make the difference in outcomes, and to rush to hospital is the best and only option.

If THR with added vasoconstriction fails, or no stabilization is obtained with the venous return filling of blood, its components, or fluids, the patient enters stage IV of impending CA.

### 2.2. Management Indoors

Indoors, if the fluid responsiveness test confirms consistent hypotension and the failure of the arteriolar system, patients should be taken to the theater and/or an interventional radiology suite for rapid source control independently of the bleeding origin.

### 2.3. Where the Bleeding Is Coming From?

Often in established hypotensive shock, there is no time for imaging or other investigations, and clinical diagnosis is the only chance of identifying beforehand the hemorrhage source.

Blood loss from arts and neck is self-evident, and so is the bleeding source identified in hospital by basic (chest and pelvis x-ray) or advanced (extended echo-FAST, iv contrast CT scan, or in selected cases, DPL or laparoscopy) diagnostic methods.

A rapid clinical assessment; extended echo-FAST for trauma to the chest and abdomen studying air or blood loss in the pleural space and blood loss in the pericardium, peri-hepatic, peri-splenic spaces, and utero-rectal or vesico-rectal pouches; basic essential X-ray to the chest, pelvis, and neck on a clinical specific basis are all common routines in any trauma center or emergency room. Neck trauma with hematoma and tracheal injury can also be detectable with an echograph.

Timing is crucial, as is a clinical assessment of the current physiological situation on impact. No diagnostic CT scan imaging or interventional angiography should delay a laparotomy for addressing and controlling the bleeding source in an unstably unstable patient with HS class 3 or 4, except when interventional radiology s needed after laparotomy for life-threatening pelvis trauma, e.g., sacral bone fractures

Hemothorax is picked up clinically with the findings of decreased *vesicular murmur* and increased percussion notes in the presence of a normal or decreased *fremitus vocalis tactilis.* Sonar is the fastest method to pick up significant intra-abdominal bleeding—diagnostic peritoneal lavage can be used where a sonar facility is not available.

Retroperitoneal bleeding can be diagnosed in absence of a CT scan and in the presence of a clinical shock picture by the exclusion of intra-abdominal hemorrhage (IAH) by echo-FAST or DPL and of thorax, limb, or neck hemorrhages.

Likewise, an intra-mediastinal closed hemorrhage from blunt injuries that amounts to a clinical picture of shock is ominous of major heart or major vessel injuries, when all other compartments have been ruled out clinically as possible sources of bleeding.

Penetrating heart injury and trans-mediastinal wounds are scenarios in which hemorrhagic shock can overlap cardiogenic shock. Cardiogenic shock (CS), being a vasoconstricting shock like HS, can mimic the latter; when, for example, a patient is found on the street collapsed with advanced shock features, no external signs of injury, and incapability to give a history of events.

Anything can be exsanguinating, from ruptured or dissecting thoracic or aortic aneurysms, massive myocardial infarction, massive pulmonary embolism, or ruptured ectopic pregnancy, when a patient is found outdoors unconscious and in shock.

When CS overlaps HS, patients usually do not reach the hospital.

## 3. Critical/In Extremis Shock, Impending Cardiac Arrest/Stroke—Stage IV

There is yet no recommendable management for critical/in extremis HS with impending cardiac arrest by exsanguination except not doing further damage and taking the patient to theatre immediately [150] [Figure 4].

Collapse in healthy subjects usually occurs at an SBP ≤ 60 mm Hg, an MAP ≤ 45 mm HG, and a DAP ≤ 40 mm Hg [151]. Patients with congestive heart failure, left main stem coronary artery stenosis, acute right heart failure, or hypertension collapse at higher values.

### 3.1. How to Manage Impending CA

Only two options can be used. The categorical aim is to avoid deterioration by breaking the subtle equilibrium maintaining the patient’s physiology.

One is the “scoop and run” tactic validated by Bickell, which can be opportunely translated in the more critical patients of stage IV shock.

In a landmark study [65], Bickell et al. conducted a landmark prospective, randomized study examining immediate versus delayed fluid resuscitation among penetrating trauma patients, which found greater survival in patients treated with delayed resuscitation compared to immediate resuscitation on the spot of the accident. Patients ≥16 years old who suffered a penetrating torso injury and had a systolic blood pressure (SBP) of 90 mmHg or lower were included in this trial. The intervention was conducted in a prehospital setting. The immediately resuscitated group received a rapid infusion of Ringer’s solution while being transferred to the hospital; fluid administration was continued at the hospital if the SBP was below 100 mmHg. The delayed resuscitation group did not receive fluids en route or when initially arriving at the hospital.

When a patient manifests signs indicating critical physiology and impending CA, the balance between life-saving reflexes is stretched to maximum, and remaining insufficient perfusion (blood, oxygen, and substrates) stays in a liable and delicate equilibrium, susceptible to any minimal change or interfering variables. Any forcing, spoiling, interrupting, or breaking of this life-holding equilibrium would push the patient to irreversible shock and *exitus*. To run towards the nearest surgical facility with no fluid/blood or exogenous vasoconstriction, leaving to the patient natural balancing mechanisms to perform at their best without iatrogenic interference until rapid/swift anesthesia/surgery for source control, is, therefore, so far, the best option presented.

This “scoop and run” option is suitable for patients in stage iv shock with no preexisting ischemic heart disease or cerebrovascular disease.

The second option, a reasonable guess and a double-edged sword, subject to blood availability, is to transfuse continuously PRC +/− plasma or whole blood; i.e., try first PRC, and if there is no positive response in consciousness, heart monitoring, or hemodynamics, add plasma, or use whole blood from start. The blood would preserve microcirculation dynamics better than any other fluid and at the same time would increase oxygen delivery by increasing the oxygen-carrying capacity.

This latter option is the one to use in patients with known ischemic heart disease or cerebrovascular disease [19]. A continuous whole blood or blood component transfusion surely will be lost down the sink out of circulation but can preserve the heart and brain from ischemic damage while passing through microcirculation and delivering its precious load.

Continuous fluid infusion in this scenario can be given together with continuous vasoconstrictor boluses or infusion. Iatrogenic vasoconstriction is meant to prevent CA. The risks of further brain and heart hypoxia during the management of such scenarios are imponderable and must be accepted.

### 3.2. Recent Tactics Used to Prevent Stage V of HS, i.e., Cardiac Arrest by Exsanguination

The resuscitative endovascular balloon occlusion of the aorta (REBOA) has a small but definite space of use and strict indication, namely, in situations far from surgical facility, a long-predicted transport time, and a proven arterial intra-abdominal or inferior limb hemorrhage.

The need for a five-minute preparatory lag for assemblage and the inevitable continuation in theatre for definitive management make the procedure restricted to the situations above, moreover as an alternative to sternotomy (about the same amount of time) and laparotomy (much less).

The real-time control of pressures and the prospect of a massive IRT have to be watched when the method is implemented [152,153,154,155,156].

The combination of an intra-aortic balloon pump and selective aortic arch perfusion with oxygenated blood recently proposed is an interesting suggestion [157]. The method, despite being more physiological and specific to the target than standard REBOA, poses the same questions of opportunity, indication, and practicability in real time as its preceding methods.

Extracorporeal cardiopulmonary resuscitation ECPR [158] has been successful in rescuing patients with cardiac arrest by heart and great thoracic vessel rupture presenting with cardiac tamponade and/or massive hemothorax [159], hypoxemic respiratory failure from lung injury, or pulmonary embolism following trauma requiring surgery or resuscitation [160].

The success of ECMO via sternotomy strongly suggests and reaffirms the crucial role of this incision in CA from the chest and, consequentially, abdominal trauma or exsanguination from the limbs.

## 4. Cardiac Arrest by Exsanguination—Stage V

Cardiac arrest (CA) by exsanguination occurs when more than 40% of the total blood volume (TBV) has been lost, usually at around half of the TBV loss. Patients then die of myocardial infarction or stroke if elderly, hypertensive, or cardiopathic, or die of insufficient venous return resulting in the heart stopping pumping. In normal subjects, heart and brain ischemia starts occurring when TBV is 40%, and CA occurs at about half of TBV loss.

Mortality is practically still close to 90% in most of the world. In Western university hospitals dedicated to trauma and advanced territory assistance, CA from exsanguination has a survival rate of around 10%, with figures between 0 and 2% for blunt trauma and 10–15% for penetrating intrathoracic trauma, up to 20% if from stab wounds [161,162,163].

Standard CPR with external cardiac massage accelerates death by increasing the speed of the remaining TBV loss.

No meaningful survival is gained by emergency resuscitative thoracotomy/thoracic aorta cross-clamping (ERT/TACC) by exsanguination from a massive intra-abdominal hemorrhage (IAH). ERT/TACC executed out of strict thoracic indications on a patient in CA only fast-tracks *exitus.*

If it is performed in a massive venous IAH from *venae cavae* or any of their major afferents (femoral, splenic, renal, iliac, portal, or retro-hepatic veins), patients will die on the spot from CA for the impairment of sufficient venous return.

If it is performed for a massive arterial IAH, the ominous ischemia-reperfusion toxemia (IRT) added by the TACC will kill the patient within hours.

The temporary clamping aiming to increase ventricle volumes and pressures is neutralized by a decreased venous return if the bleeding source is venous or by conspicuously worsening the already-present IRT if the bleeding source is arterial.

Even in audits of experienced units with high trauma intake and a high ERT/TACC overall immediate survival [164], in the subgroup of TACC performed for massive IAH, survival in the ICU longer than 24 h is dismal, as many perish within 4 to 16 h at maximum.

A crush laparotomy in lieu of a thoracotomy has much higher odds of survival for the simple fact that it avoids a physiologically exhausting, energy-expensive, and time-consuming procedure such as an ERT. An experienced team of two nursing staff can reach and deal with massive ITH or IAH within minutes, one for ingress and two or three for source control, often within a maximum of 45 min in all sites, notable exceptions being bleeding from pelvic, retro-hepatic, and pulmonary veins.

TACC for massive intra-abdominal hemorrhage in a patient not in CA is only to be condemned. TACC survivors are to be found only among patients with intra-thoracic bleeding [163,165].

Once CA from exsanguination sets in, which means the patient has lost half of their TBV, only a war plan in situ can rescue them from death. Once the patient becomes pulseless, the window of opportunity is very brief as the critical warm ischemia time is only 5 min or less for the brain and about 10 min for the heart [166,167].

The brain cannot tolerate in a cardiac arrest a warm ischemic time longer than 4–5 min in an adult [166,167]; the heart is up around 10 min. A limbo period of a further 10 min of rescue feasibility has been noted for healthy hearts. Following cardiac arrest, the brain, heart, and the rest of the body organs enter a state of limbo known as “downtime”. This is the twilight zone, in which the process of dying begins. The downtime period within which a healthy heart can be restored to rhythm is no longer than 20–30 min [168].

CAs up to a 30 min duration from a penetrating intra-thoracic injury have been rescued in oxygenated patients by emergency resuscitation thoracotomy (ERT) [169].

Only intubated or otherwise oxygenated patients arriving with witnessed respiratory cardiac arrest no longer than 5–10 min in duration should undergo resuscitation in blunt chest injuries [170].

If the scene is beyond the ten-minute window and the CA is witnessed on the spot, ERT should be performed in situ, if possible. ERT is indicated in a CA from blunt trauma who arrives within 10 min from a witnessed CA if intubated or 5 min if not intubated; otherwise, this is a futile exercise with a survival rate of 02% in blunt trauma [161,162,163,171]. If at any time the rhythm changes to asystole, is asystole from the beginning or remains slow (< 40 beats per minute), if the patient is in PEA, if a central pulse is not palpable, or if there is a systolic inferior to 70 on the peripheral pulse, i.e., not detectable, then resuscitation can be given up in situ. If the patient gains ROSC, they can be brought to the ER [171].

Criteria for the termination of ERT have been authoritatively established [172].

In penetrating chest injuries with signs of life on the scene, resuscitation via ERT is still possible at 30 min from cardiac arrest with 10–15% survival, up to 20% with stab wounds [161,162,163,169,171].

An open approach to cardiac arrest following trauma or not traumatic hemorrhage is a must.

External chest/cardiac massage or compressions only hasten *exitus* by increasing stroke volume and bleeding speed and worsening mechanical damage. No CA, whether from exsanguination or from major great thoracic or abdominal vessel disruption, should receive standard medical CPR. It would only accelerate *exitus*. Only by rapid transport indoors within 10 min for an ERT +/− damage control anesthesia and surgery can the patient survive. Whether to administer or not a GA in a patient exsanguinated in CA is an ongoing debate, due to the scarce knowledge we have on the limbo period.

Closed external or open internal CPR can be effective only after the solution of continuity has been controlled and there is enough Pmif to make CPR meaningful and successful [173].

ERT/TACC is lifesaving in exsanguinations and cardiac arrest from intra-thoracic bleeding or heart/hilar life-threatening injury.

The rules are simple. (1) Go for the abdomen if the bleeding is intra-abdominal and for the thorax if it is intra-thoracic—the abdominal aorta and the aortic arch are clamped as well or better. (2) No cardiac arrest by exsanguination can be reversed if first the solutions of continuity in the vascular system are not controlled and venous return is not meaningfully restored sufficiently to allow the heart to restart pumping again. (3) Never use TACC before plugging and transfusions.

The stand-by use of vasoconstrictors may prevent, delay, or sometimes reverse a cardiac arrest. A lion bite at the base of the neck or a transverse cut soon below at the inguinal fold transecting both femoral vessels kill within 5–10 min.

A rapid emergency resuscitative anterolateral thoracotomy or sternotomy allows source control, manual massage, and intra-operative venous return rapid restoration by a pre-inserted central venous line (CVL) or by direct in vivo superior vena cava (SVC)/right atrial cannulation.

## 5. The Essential Damage Control Resuscitation (Damage Control Anesthesia and Damage Control Surgery)

### 5.1. Titrated-to-Response Anesthesia

The aim of general anesthesia for hemorrhagic shock is to prevent or worsen hypotension and hypoxemia, normalize blood pressure and oxygen, and not to worsen or reverse unfavorable detrimental corporeal temperatures [174].

Severe hypotension during intubation in a patient with HS can lead to cardiac arrest and death. Mechanisms that contribute to peri-intubation hypotension include vasodilation from induction medications, decreased sympathetic tone from sedation, and decreased venous return from increased intra-thoracic pressure with positive pressure ventilation.

Pre-existing hypotension prior to intubation is often counteracted by administering blood/blood products for hemorrhagic shock, intravenous fluids, and timely used vasopressors. For patients at risk of hypotension during intubation [175], induction agents contributing to hypotension must be avoided [176].

Skillful anesthesia for surgery in patients with critical physiology is essential for patient survival.

Satisfactory and safe general anesthesia with or without airway protection can be performed under total intravenous anesthesia (TIVA) with oxygen and ketamine or a synthetic short-action opioid-like remifentanil, a drug with the shortest half-time and emergence time.

Ketamine is excellent for induction and maintenance, no less so as an analgesic and sedative, but increases oxygen consumption (VO_2_) and heart rate in an organ, the heart, which is dependent on flow to increase oxygen delivery, and at basic regimen, it extracts 75% of the oxygen delivered (DO_2_). Therefore, it should be used only in healthy patients. In HS, it can be used for interventions on compensated (mild to moderate) shock on normal or healthy but not coronaropathic patients with a heart rate not higher than 120 bpm. Ketamine is the only anesthetic agent that does not depress the myocardium, the vasomotor tone, and the airway tone as well. For these reasons, it has been successfully used in war-outdoors scenarios as an induction agent [177,178] or for TIVA in patients not in HS spontaneously breathing air oxygen only [179].

Diazepam can be given after airway and source control with a normalized blood pressure to buffer the neurotropic, extrapyramidal, and chronotropic effects of ketamine, while a glycopyrrolate bolus controls the increased salivation without crossing the blood–brain barrier or increasing the heart rate. It is the fascia, the membrane, and the skin that are sensitive to pain and increase autonomic reflexes and tachycardia. Succinylcholine deals rapidly with the rare laryngospasm caused by ketamine or by remifentanil [179].

Oxygen and alfentanil, the fastest-acting synthetic opioid, or oxygen and etomidate [176,180], are best suited for induction in cardiopathic patients, due to their efficacy and the scarce interference with cardiovascular functions. The new (L) isomer of ketamine, S (+)-ketamine, promises, in virtue of lesser chronotropic and neurotropic effects as compared to the classical racemic mixture, to become an invaluable drug in anesthesia for critical illness, extending ketamine usage in cardiopathic patients [181,182,183,184].

Propofol is hypotensive and suits well only mild or moderate compensated hemorrhagic shock with TBI, due to its property of decreasing endocrinal pressures.

In advanced stages, etomidate and ketamine, though preferable to propofol in HS, may well be not convenient choices, one for its action on adrenals, and the other for its tachycardic effect.

In such critical patients, induction must be performed with oxygen and a synthetic opioid. The more unstable or critical the hemodynamics of the patient is, the more the simple and crude combination of oxygen and a synthetic opioid should be used for induction and maintenance.

GA maintenance can then be performed with “TIVA on demand” keeping in mind the pain-sensitive structures encountered during the different body ingress sites. Only skin incision and fascia incision and stretching, such as serosa and capsules, trigger pain and deserve a sufficient level of circulating anesthetic. The era of flat anesthetics and surgery is over, and it is time we go toward a tailored-to-individual physiology restoration in real time. Damage control anesthesia, or autonomic-controlled anesthesia, can be safely and satisfactorily used under total intravenous anesthesia (TIVA) with oxygen and ketamine or a synthetic short-action opioid-like remifentanil, a drug with the shortest half-time and emergence time, with or without airway protection.

Oxygen and alfentanil, the fastest-acting synthetic opioid, or oxygen and etomidate, are best suited for induction in HS or cardiopathic patients, due to their scarce interference with cardiovascular functions.

In the combined scenario of TBI and HS, continuous intravenous anesthesia (CIVA) with propofol and ketamine infusions offers the control of endocranial hypertension and systemic hypotension. Arterial monitoring and an intracranial probe are necessary if trunk surgery is performed first

### 5.2. Damage Control Surgery

The term ‘damage control surgery’ (DCS), though applied unwittingly in acute surgery and not traumatic hemorrhage for a decade, and in war and elective major surgery much earlier at the beginning of the last century, was popularized in the surgical management of trauma with hemorrhage in the 1990s [185,186,187]. It describes conduct of “dealing only with the damage at the impact operation”, referring to the completion of the procedure at a second operation after the physiological stabilization or improvement of the patient. Eventually, the term was extended to non-trauma emergency surgery, e.g., severe peritonitis [188].

The great majority of surgical interventions have generally two phases: (1) an etiology/source-targeted phase, with the arrest or control of a hemorrhage and the removal of contaminated, infected, necrotic, gangrenous, ischemic tissues or fluids; and (2) a reconstructive, restorative primary healing phase.

In some surgeries, the demolition phase prevails, e.g., neurosurgery or thoracic surgery; in others, there is a prevalence of a pure reconstructive phase, e.g., vascular, plastic, orthopedic, maxillofacial, or alimentary tract.

In some circumstances of trauma and emergency surgery, in the presence of severely deranged physiology, the two phases are not executed at the same initial operation, impact, or index operation, and the reconstructive phase is safely postponed.

All acute care surgery is aimed to prevent death from hemorrhagic shock (HS) or from inflammatory (toxic and septic) shock (IS). ‘Source control’, i.e., arresting a progressing hemorrhage, and ‘source removal’, i.e., the elimination of contaminated, infected, and necrotic/ischemic/gangrenous tissues and fluids, are the main stem rationales and reasons why we intervene in the same time end-point landmarks of competence and efficacy. This is what is meant by damage control surgery (DCS).

The DCS concept was initially promoted and motivated for the wrong reasons, i.e., to prevent the establishment and advance of a supposed ‘lethal triad’ (acidosis, coagulopathy, and hypothermia) [187], whereas the only lethal factor is, in fact, acidosis! Coagulopathy does not affect mortality in trauma [137]; metabolic hypothermia is a function of the degree of cell hypoxia and not a cause; moreover, it does not affect mortality in trauma. The concept of the lethal triad in trauma in the last four decades has misled clinical research on trauma and derailed surgeons from understanding the real crunch of the DCS tactics.

#### 5.2.1. Metabolic Acidosis

Metabolic acidosis (acid pH, excess of lactate, and negative base excess) is universally present in shock as a direct consequence of low perfusion, tissue ischemia, and hypoxia [189]. Moreover, acidosis dampens catecholamine response during resuscitation, impairs healing, and shifts the Hb dissociation curve. Pre-existent or concomitant liver disease, alcohol consumption, diabetes mellitus, and chronic obstructive pulmonary disease are potential pitfalls when interpreting an increase in lactic acid levels. pH is slow to adjust variations, making negative base excess (NBE) the best and most reliable real-time parameter of metabolic acidosis [190,191]. We know with certainty that persisting pH <7.1 and a lactate >15–16 mmol/l are invariably lethal for impairing basic intracellular metabolic processes [192] and that NBE >6–8 mmol/L is a sign of severe hemodynamic/metabolic derangement [191].

ScvO2, another reliable marker, is, though, the most synchronous parameter of cellular hypoxia. It can be valued in both central and peripheral venous blood. It should be used in the assessment of the severity of a hemorrhagic shock, in association with NBE, as a reliable warning sign to permit or avoid the reconstructive phase at impact operation.

#### 5.2.2. Criteria for DCS

The criteria for DCS in trauma surgery were indicated at the introduction of the terminology in the surgical literature [187] and are continuously put under re-evaluation [193,194,195]. All indications for damage control surgery proposed so far come out as a changing joust, a tilt of indications and numbers. On reading, one may become lost in translation! What is the common denominator [196]?

A more synoptic, intuitive, factual, and scientifically solid ensemble of criteria should be used, where common denominators are rapidly identifiable and snapshot-easy to apply [197] [Figure 5].

The time to source control, refractory hypotension, advanced tissue hypoxia/metabolic acidosis, and intra-operative problems remain the main stem criteria for applying DCS in an emergency or trauma surgery. Other specific criteria can be extrapolated by experience: specifically, a delay to source control [144,198], preoperative hypotension [199,200,201,202], low Hb or an Hb drop at the first sample [203], a preoperative or intra-operative loss of 1.5 or 2 L of TBV [204,205], and intra-operative hypotension, especially if TBI is also present [202].

The studies of Asensio et al. and Ordonez et al. [206,207] point out further insights and confirm two facts. With 30% of TBV loss, around 1.5 L in a 70 kg average patient, hypotension is unexceptionally present, whereas we know by experience that with lower TBV losses at a range of 20–30% of TBV, hypotension is inconstant and may or may not have set in as yet. Indirect proof that consistent hypotension with a 30% TBV loss is a crucial pivot point in hemodynamic homeostasis in less urgent scenarios like one of a hemothorax losing 1.5 L of blood within 24 h where thoracotomy is often required [206,207].

A 2 L loss of blood is, in an average adult, around 40% of TBV, which is an ominous sign of critical physiology. In a pregnant person, a 35% TBV loss is a cutoff for hypotension, and at the same time an ominous sign of severe derangement that is about to become critical and requires the most rapid intervention. CA by exsanguination occurs generally at levels of half of TBV loss, lesser in cardiopathic patients [1].

#### 5.2.3. The Real Rationale for Performing DCS

What decreases early mortality in trauma with hemorrhage is the earliest intervention of source control.

What decreases midterm morbidity and consequent mortality in trauma with hemorrhage is the prevention or buffering of the invariably occurring ischemia-reperfusion toxemia (IRT), causing early healing failure (abdominal wall, anastomosis) or a late second hit of MODS/MOF. IRT determines the second hit peak of morbidity and mortality (MODS/MOF), augmented by the breakage of the gut-barrier function of the intestinal mucosa secondary to ischemia, triggering further IRT, bacteremia, endotoxemia, and eventually systemic inflammatory response (SIR) [20,21,22,23]. The natural outcome of these reverberating and self-amplifying pathways is MODS/MOF and death [24,25].

Orthopedic surgeons recognized the relevance of post-traumatic inflammatory response since the 1980s when faced with the dilemma of the timing of fracture fixation in a polytrauma with multiple fractures [208]. It was concluded that fixing the fractures early was the best way to prevent or reduce the post-traumatic inflammatory response inevitably occurring in such injuries, especially in blunt ones without hemorrhage.

In a scenario of blunt poly-trauma, without shock, IRT and SIR often coexist as synchronous effects of ischemia, soft tissue injury, and inner clot contamination/infection.

In a trauma with hemorrhage, SIR has a minor role, as the inflammatory cascade becomes lost with the blood loss.

In fact, in main orthopedic poly-trauma without HS, SIR causes an early second hit, if no early care is performed for major fractures and soft tissues [209,210,211,212]. Likewise, in blunt polytrauma, with hemorrhage, IRT and SIR often coexist, due to the synchronous effects of ischemia, soft tissue injury, and inner clot contamination/infection, with IRT as an early phenomenon and SIR as a late one, whereas in penetrating trauma with hemorrhage, only IRT can follow as an early second hit, an effect of hypoxemia with ischemia, as the actors of the inflammatory cascade become lost out of circulation.

Secondary SIR would occur only if the causes and effects of the IRT are not dealt with by source control and source elimination, triggering the lethal cascade leading to MODS/MOF and exitus.

Post-traumatic inflammatory response or systemic inflammatory response (PTIR/SIR) occur immediately in a blunt trauma with scarce or no hemorrhage. It can kill a patient on the spot if massive, even before cardiovascular accidents or massive hemorrhages take their toll. It occurs late in infected hematoma [122,123].

IRT occurs early in trauma with hemorrhage or in the presence of a toxic shock.

Both IRT and SIR cause the second hit (MODS/MOF) with the involvement of the kidneys and, more ominously, the lungs (secondary acute lung injury and dysoxic respiratory failure). A lag period of gradual, not subsiding, relentless, silent, subclinical shock with the underneath microcirculation striving despite normal macro-circulation variables, precedes the second hit [20,21,22,23,24,25].

It is clear that to state DCS has reduced the mortality of trauma with hemorrhagic shock because it prevents the setting in or worsening of the lethal triad is a misleading statement that has distracted research on HS for decades. It is the avoidance of the second restorative, reconstructive primary healing, with a high-to-very high risk of healing failure (abdominal wall, intestinal anastomosis, soft tissues cover) in presence of a severely deranged physiology, that avoids or prevents or buffers the entity of complications and their related mortality [213].

The effects of deranged microcirculation, hypo-perfusion, and hypoxemia on healing have long been known [214,215,216,217,218]. Further risk is added in ICU when often shocked patients are put on catecholamines [219,220]. Using staples for anastomosis in the reconstructive phase may decrease the risk of late anastomotic failure [221].

Exactly there, it is where the window of improved survival would be found, if all retrospective studies on deaths and complications were aimed at finding and correlating the influence of IRT and SIR on healing failure. Retrospective studies should compare morbidity, mortality, and hospital stay times between two groups of trauma patients with analogue or similar physiological and pathological statuses, one of which was treated with DCS and the other with primary reconstruction. As a matter of fact, the benefits of DCS on survival have not been shown as yet [222,223,224,225,226,227,228,229,230]. The strategy, however, must bring benefits and is accepted in current practice.

The reason for this is the distraction brought by the lethal triad, whereas, instead, the crunch of the damage control surgery concept is in the energy failure of anastomosis and abdominal wall healing, contraindicating primary restorative procedures, due to the effects of IRT, occult hypo-perfusion and MODS on the anabolic processes. An analysis of deaths from severe trauma would have hinted to healing failure as the primary event progressing to *exitus*. The only way to show the benefits of DCS is through retrospective studies comparing groups of patients treated with primary reconstruction and physiologically similar groups managed with DCS [213,231].

## 6. Conclusions

The heuristic approach is fundamental.

A shift in approach from ‘clinics trying to fit physiology’ to one of ‘physiology to clinics’, and management exclusively based on the modulation of physiology, is the optimal way to tackle and study progressing HS.

There is a liability and weakness in most studies in the mental approach attempting to work using a top-to-bottom perspective, especially in a topic where personnel and unit/center/hospital exposure and expertise vary from place to place, and from operator to operator invariably, too. All meta-analysis presents the same basic bias: to pretend to see from the tips of the icebergs what happens below the surface of the sea and the icebergs’ bases. Correlation or association does not necessarily imply causation! A deeper and more causative factor, or pure causality and coincidence, can be at the base, the reason for a correlation, if a plausible direct cause–effect relationship is not first proven [232].

*“The probability that a research claim is true may depend on study power and bias, the number of other studies on the same question, and, importantly, the ratio of true to no relationships among the relationships probed in each scientific field. A research finding is less likely to be true if the studies conducted in a field are smaller, the effect sizes are smaller, there is a greater number and lesser preselection of tested relationships, there is greater flexibility in designs, definitions, outcomes, and analytical modes, and there is greater financial, other interest and prejudice. For many current scientific fields, claimed research findings may often be simply accurate measures of the prevailing bias”*.[233]

Of the two, one: either we accept the established and observed hemodynamic laws as absolute, or we purposefully neglect and discard them and believe, for example, that coagulation, which does not minimally relate such laws, can affect bleeding flows and mortality in HS. Empiric truth remains our intelligible base and aim and, as common practice indicates, it tells a more solid story about bleeding. That is, HS outcomes depend only and solely on the main stem flow of facts and phases: (1) not worsening, stabilizing, or improving patient physiology before source control, (2) timely, earliest source control, (3) blood and oxygen replenishment, and (4) manipulations of oxygen and temperature during phases (1) and (3). Nil else can affect mortality.

The mortality of hypotensive shock of about 4050% and 90% in the groups of stages IV and V has not changed since some medical pioneers started describing and studying hemorrhagic shock [234,235,236,237]. Their insights are still actual.

There are only three ways to decrease the mortality and morbidity of a salvageable HS: to know reliably and in real time the cardiac circulatory reserve of a bleeding patient and the time-lapse of the remaining and diminishing TBV before reaching critical volume and pressure; to know rapidly if the bleeding is arterial or venous or a mix, and the sites of origin; to monitor microcirculation function, independently from macrocirculation normalization. There is where research should go. Almost no studies keep in account the body’s compensatory capacity affected by the arteriolar gate system or the physiological response in general.

The physiological classification of HS is, so far, the most sound and pragmatic stand-by in situ evaluation of hemorrhagic shock dynamics, easy to apply, scientifically motivated, and useful to define the timing for THR and the timing for DCS application.

## Figures and Tables

**Figure 1 jcm-12-00260-f001:**
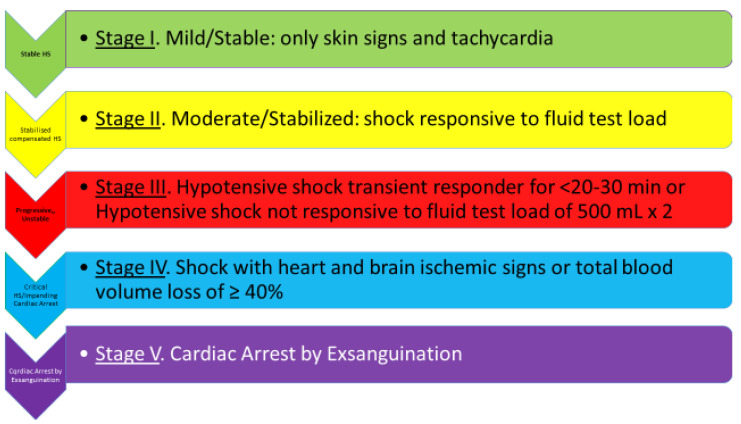
Diagnostic classification of hemorrhagic shock.

**Figure 2 jcm-12-00260-f002:**
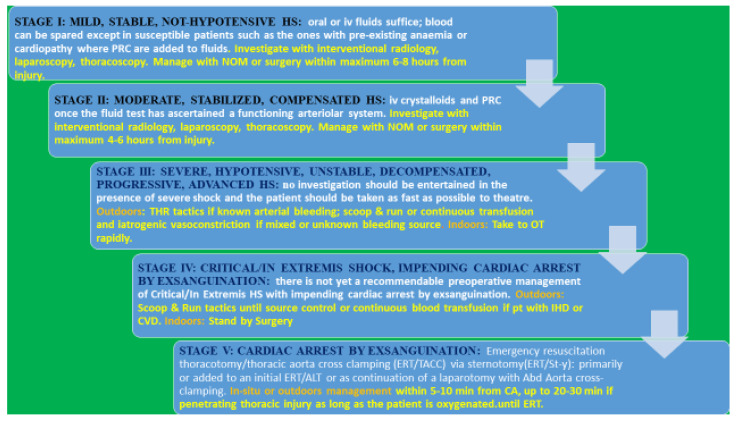
Therapeutical classification of hemorrhagic shock. NOM = non-operative management; PRC = packed red cells; OT = operating theatre; CA = cardiac; ARREST St-Y = sternotomy; TACC = thoracic aorta cross-clamping; HR = hypotensive resuscitation; ERT = emergency resuscitative thoracotomy; ALT = anterolateral thoracotomy; THR = titrated hypotensive resuscitation.

**Figure 3 jcm-12-00260-f003:**
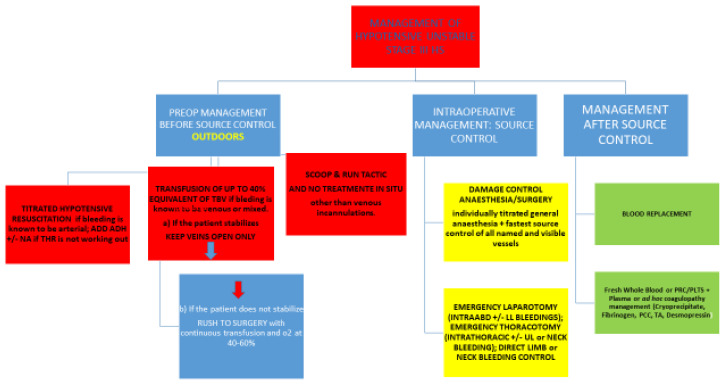
Management of stage III HS. TBV = total blood volume; CVL = central venous line; PRC = packed red cells; PLTS = platelets; PCC = prothrombin complex concentrates; TA = tranexamic acid; HR = hypotensive resuscitation; ADH = anti-diuretic hormone; NA = noradrenaline; LL = lower limbs; UL = upper limbs.

**Figure 4 jcm-12-00260-f004:**
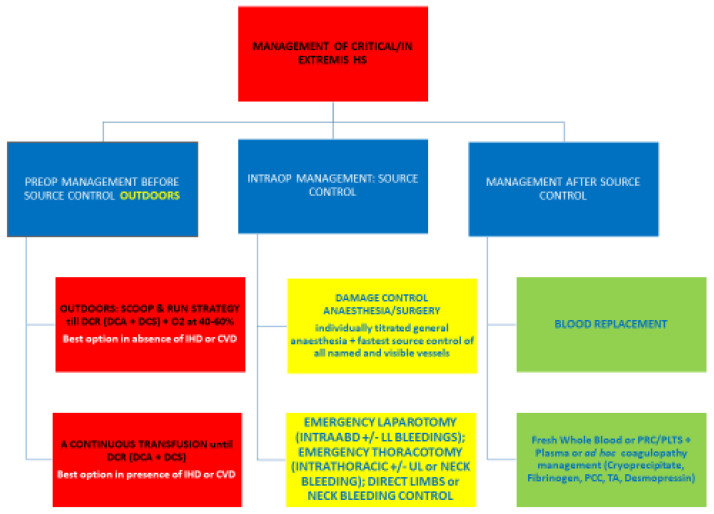
Management of stage IV HS. DCR = damage control resuscitation; DCA = damage control anesthesia; DCS = damage control; SURGERY LL = lower limbs; UL = upper limbs; PRC = packed red cells, PLTS = platelets; PCC = prothrombin complex concentrates; TA = tranexamic acid, IHD = ischemic heart disease; CVD = cerebrovascular disease.

**Figure 5 jcm-12-00260-f005:**
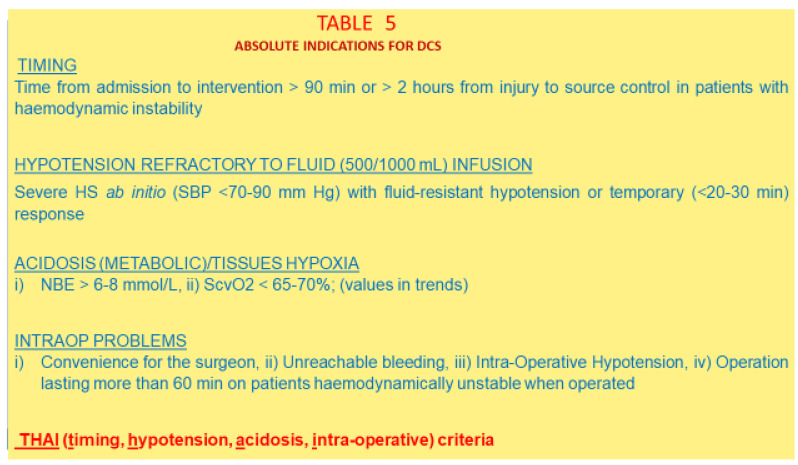
Criteria for damage control surgery. DCS = damage control surgery; SBP= systolic blood pressure; NBE = negative base excess; ScvO2 =central venous oxygen saturation.

## Data Availability

Not applicable.

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
