# Peer review of "Management of Hemorrhagic Shock: Physiology Approach, Timing and Strategies"

_jcm, 2022, doi:10.3390/jcm12010260_

Round 1

Reviewer 1 Report

The paper is an interesting and exhaustive narrative review: 

It is well structured and organized, clear in all its steps

I think figures 2, 3, and figure4 should be more evident and improved: with these colors, they are not clear and well done

Author Response

To Reviwere 1

Thanks very much for your observations, suggestions and critique.

Indeed the Tables had to be redone. I hope I did a good job in improving colours and more importantly clarity and legibility. Tables 1 & 2 were improved in colours and presentation. Tables 3 & 4 had to be redone and could not make better than I tried - see below. Consequently I had to review the related parts of the text, bring few changes without changing or modifying the almost totality of the manuscript. I am easy and prepared to accept and improve the presentation according to your second review.

Thanks for your time and contribution to the work.

Faithfully

Dr F Bonanno, 22.11.2022

Reviewer 2 Report

Management of trauma haemorrhage has been an active area of research for decades, yet mortality remains unacceptably high. Recent RCTs of haemostatic resuscitation comparing blood components or whole blood to fluids have not significantly impacted survival in this heterogeneous patient population. This review provides a refreshing read that cuts across the reductionist findings of the majority of these RCTs, by identifying and matching the changing physiological status of the individual to relevant treatments that may make a difference to systemic outcomes and survival. This is a real-world approach to trauma and haemorrhagic shock, instead of the often one-dimensional approach of many RCTs with interventions that are physiologically disconnected from their pragmatic outcome measures. The author correctly observes that many interventions, if not matched to the precise physiological status of the patient, do not significantly improve survival in the real world.

This review is comprehensive, up to date and authoritative. The content is excellent and follows a logical progression throughout the manuscript. Overall, this manuscript reflects the accumulated knowledge and experience of this author. The text is very long, and some effort should be made to reduce length, primarily via reduction of repetition and reiteration of personal opinion. However, the topic deserves the extensive discussion presented here, and manuscript length is not a critique in this instance.

Use of abbreviations could be standardised to improve readability.

The manuscript requires moderate editing for language and style, but in my opinion, the editorial team could perform this.

The figures provide an adequate summary of the main points, but the choice of colour and font make some difficult to read, particularly Figs 2, 3, 4.

Author Response

To Reviewer 2

Thanks very much for your observations, suggestions and critique.

Indeed the tables had to be redone. I hope I did a hood job in improving colours and more importantly clarity and legibility. Tables 1 & 2 were improved in colours and presentation.

Tables 3 & 4 had to be redone and could not make better than I tried - see below. Consequently I had to review the related parts of tye text, bring few changes without changing or modifying the almost totality of the manuscript.

I am easy and prepared to accept and improve the presentation according to your second review.

Thanks for your time and contribution to the work.

Faithfully

Dr F Bonanno, 22.11.2022

Reviewer 3 Report

Overall this is a nice review of management of hemorrhagic shock. It addresses various components of care for these critically ill patients. There are a number of concerns that should be addressed prior to publication. Most concerns are related to outdated references. Primarily use of colloids (section 280-296) have largely been abandoned and claims made in this section have been contraindicated. Would include more detail on early use of blood products in conjunction with crystalloid minimalization and information on massive transfusion protocol. Would clearly define categories of fluid responsiveness earlier in the review. The timeline of up to 30 minutes to assess responsiveness seems very generous. 

There has also been recent evidence for prehospital use of plasma, whole blood and TXA in bleeding trauma patients (see PAMPER Trial and Crash-2 for example). Would also elaborate on use of TTE in trauma resuscitation in section 88-90 with citations. 

To make the manuscript more accessible to readers the number and use of abbreviations should be reduced. One example that often leads to confusion is HR, for hypotensive resuscitation, which can be mistaken for heart rate, would use the alternative term permissive hypotension. 

Figure 1 and 3 are too small to read/have low resolution. 

Author Response

To Reviewer 3

Thanks for you spotted and ameliorative review!

It has taken 5 full days to comply with your suggestions, observations and critique.....and 6-8 months of spare not dedicated time, except in the last couple on months, to write the work.

I have received and in-taken all but part of the review was probably mislead by some small presentation errors of mine in the original submission.

As a matter of fact I did quote the studies. I did not comment more only because I am very skeptical on 'coagulopathy in trauma' and wanted to stress the title key words more than specific treatments. So I have handled the topic with 'fine forceps', making sure I do not crash against the currently prevailing views.

I have readjusted the first 2 Tables aesthetically by trying to match colours in the best way but focusing on the legibility and clarity. I can only hope that from external observer I have improved their legibility. I am easy and prepared in having suggestions or specific observations on aesthetic.

Indeed Tables 3 and 4 needed yo be re-done. In order to make clear one of the key messages of the manuscript, I had to review carefully the related text as well. So I made it more clear and legible, eliminated repeats, and as result I had to add few extra paragraphs. Consequently I also reviewed appropriately the related references.   

I am looking forward to have a review on my due changes, as triggered by your first review.

Thanks for your time and contribution to the article.

Faithfully

Dr F. Bonanno, 22.11.2022 
